# *Alhagi sparsifolia* Harbors a Different Root-Associated Mycobiome during Different Development Stages

**DOI:** 10.3390/microorganisms10122376

**Published:** 2022-11-30

**Authors:** Zhihao Zhang, Xutian Chai, Yanju Gao, Bo Zhang, Yan Lu, Yi Du, Yulin Zhang, Ya Ding, Akash Tariq, Abd Ullah, Xiangyi Li, Fanjiang Zeng

**Affiliations:** 1Xinjiang Key Laboratory of Desert Plant Roots Ecology and Vegetation Restoration, Xinjiang Institute of Ecology and Geography, Chinese Academy of Sciences, Urumqi 830011, China; 2State Key Laboratory of Desert and Oasis Ecology, Xinjiang Institute of Ecology and Geography, Chinese Academy of Sciences, Urumqi 830011, China; 3Cele National Station of Observation and Research for Desert-Grassland Ecosystems, Cele 848300, China; 4University of Chinese Academy of Sciences, Beijing 100049, China; 5College of Resource and Environment Sciences, Xinjiang University, Urumqi 830046, China

**Keywords:** desert plants, internal transcribed spacer, root-associated fungi, *Alhagi sparsifolia* Shap., co-occurrence network

## Abstract

The mycobiome in the rhizosphere and within the roots benefits the nutrition and function of host plants. However, compared with the bacterial community, root-associated mycobiomes of desert plants and the forces that drive their assemblage are limited. Here, we investigated the mycobiomes in bulk soil, rhizosphere, and root compartments of *Alhagi sparsifolia* Shap., a phreatophyte species dominating in Central Asia. The internal transcribed spacer (ITS) gene phylogenetic profiles displayed significantly diverse mycobiomes across three compartments and host growth times, together explaining 31.45% of the variation in the community composition. The community structure of the perennial stage was markedly different from that of other stages (30 days to 2 years old). Along the soil–plant continuum, the α-diversity (estimated by Chao1) decreased gradually, while concomitantly increasing the community dissimilarity and the influence of edaphic factors. Specific leaf area, soil water content, and soil organic matter levels were common factors driving the composition of the three mycobiome communities. A more complex and connected network was observed in the root community compared with the other compartments. Overall, our work suggests that an age-sensitive host effect restructured the desert-plant-root-associated mycobiome, and that edaphic factors and host growth strategy may play potential roles in this process.

## 1. Introduction

A variety of fungal species associated with plant roots, including beneficial (e.g., mycorrhizal fungi) and harmful fungi (e.g., plant pathogens), can affect the survival, competitiveness, and tolerance to biotic and abiotic stresses of their hosts [1,2]. The exploration of the plant–microbe interactions is of special interest as it can aid in the elucidation of their roles in plant growth and development, as well as apply their interactions to promote plant productivity and phytoremediation [3]. The driving factors and mechanisms of the microbiome assembly have been extensively studied; however, these works primarily focused on bacterial or rhizosphere samples [4,5]. To date, little is known about the detailed characterization of mycobiome, including diversity, community composition, and co-occurrence patterns [6], especially in the spatial resolution of distinguished root-associated compartments (i.e., bulk soil, rhizosphere, rhizoplane, and root) of desert plants.

Deserts cover about one-third of the world’s land surface [7]. The extreme environmental conditions of desert ecosystems, such as drought, oligotrophy, solar irradiation, high salinity, intense wind erosion, and environmental physical instability, place additional pressure on local plants, resulting in a desert landscape dominated by sparse shrubs and grasses [7,8]. Although deserts represent many serious environmental challenges, plants and microbes have developed adaptive mechanisms that allow them to grow, survive, and reproduce in these extreme conditions. Indeed, plants and microbes have been treated as halobionts rather than standalone entities [5]. For example, our recent study revealed a significant correlation between the composition of the rhizosphere mycobiome and proline accumulation and superoxide dismutase (SOD) activity in leaves of *Alhagi sparsifolia* Shap., a perennial leguminous species [9]. Root-associated microbes drive the plant uptake of nutrients, including nitrogen (N) and phosphorus (P) [10]. Previous studies suggest that the adaptation of desert plants is partially due to their symbiotic relationship with microorganisms. However, little is known about the fungal species associated with desert plants, the communities they form, and the forces that influence them [11].

The surrounding edaphic properties and the host plant can exert a dynamic influence on the root-associated communities. In bulk soil, some specific microbial taxa can be selectively recruited by the root exudates of host plants to form continuous soil–plant communities [12]. In addition, plants can secrete bioactive molecules into the rhizosphere to alter the soil environment for microbial activity [13]. The attributes of root exudates are affected by host growth and development; hence, the assembly of the root-associated microbiome is driven by both dynamic root exudates and the substrate preference of microorganisms [14,15]. A recent study revealed that plant functional traits (e.g., specific leaf area [SLA]) can predict unique components of soil fungal diversity and community composition [16]. However, these variations cannot be explained by abiotic soil properties or climate. Therefore, deciphering the variations in root-associated fungal communities at different growth and development stages of desert plants can help to comprehensively understand the coevolution between desert plants and microorganisms.

Plants growing in the Taklimakan Desert are dominated by perennial phreatophytes with roots that can reach deep into the groundwater. The huge potential evaporation and negligible precipitation make it difficult for these plants to reproduce via seed [17]. Although the type of plants can germinate using irrigation provided by the sporadic seasonal flood, the receding flood ultimately kills the seedlings. As a result, the seedling stage is rare in the natural conditions of this region. Therefore, clonal reproduction is more common in these perennial phreatophytes [18]. *A. sparsifolia* is a typical phreatophyte that has also been used in stabilizing dunes, high-quality forage, and Uygur medicine [19,20]. The plant produces seeds once it reaches 3 years old.

Here we investigated the temporal dynamics of the mycobiome associated with *A. sparsifolia* roots and their driving factors through the combination of the pot (for its seedlings) and field (for its perennial) experiments. The root-associated communities were focused on three belowground compartments, bulk soil, rhizosphere, and root via high-throughput sequencing. These mycobiomes were investigated at different growth and development stages using 30, 60, 90-day, and 2-year-old potted *A. sparsifolia* seedlings and naturally occurring perennials (>3-year-old). Based on previous findings of other plant species (e.g., wheat, *Arabidopsis thaliana*, etc.), we proposed two hypotheses: (1) significant variations are present in the root-associated mycobiome, including species diversity, community composition, and co-occurrence pattern at different growth times of the host, and (2) the host effect and soil physical-chemical properties jointly drive the variation of the root-associated mycobiome. This study, therefore, aims to provide an in-depth mechanical understanding of desert-plant-microbe interactions and implications for desert vegetation management.

## 2. Materials and Methods

### 2.1. Study Site and Experimental Design

Our experiment was carried out at the Cele National Station of Observation and Research for Desert-Grassland Ecosystem (80°43′45′′ E, 37°00′57′′ N) in Xinjiang, northwest China. This study site is located on the southern fringe of the Taklamakan Desert, Central Asia, which is characterized by a hyper-arid climate [21]. In the natural habitat, the low water availability in the topsoil prevents the germination of *A. sparsifolia* seeds except under irrigation by seasonal floods [18]. Thus, we designed in situ potted experiments to simulate flood events by irrigating groundwater to reproduce *A. sparsifolia* via seed.

*Alhagi sparsifolia* seeds were collected from their natural distribution area (36°17′–39°30′ N, 80°03′–82°10′ E) in 2018. In the pot experiment, topsoil (0–30 cm) in the natural community of *A. sparsifolia* was selected to minimize the soil condition difference between the potted seedlings and the natural plants. Ten seeds were sown 3–5 cm deep in a 90 L pot and were watered with 1.5 L groundwater every 2 days to ensure the optimal growth of seedlings. Such irrigation events continued until the end of the growing season (September) and continued at the beginning of the next growing season (May). After 30 days following emergence from soil, only one seedling was left in each pot. Seedling samples, including leaves and roots, bulk, and rhizosphere soil, were collected at 30 (30 d), 60 (60 d), 90 (90 d), and 455 (2 a) days after emergence. In the nearby desert–oasis ecotone, the perennial *A. sparsifolia* plants (~86 cm height) were selected on 31 August 2020. All treatments were carried out in triplicate.

### 2.2. Sample Collection

The fungal samples along the soil–plant continuum were collected according to the description by Chen et al. [22], including bulk soil, rhizosphere soil, and roots. Briefly, the soil that was loosely attached to the roots was used as bulk soil. The fine roots and their closely attached soil were placed in a sterile centrifuge tube and then shaken with a vortex oscillator. The soil at the bottom of the tube was then designated as the rhizosphere soil. Subsequently, the roots were moved into another tube and washed with sterile water. The soil that was washed away was excluded from the rhizosphere soil. The washed roots were used as root samples. However, this methodology of the separation of root compartments did not distinguish the microhabitats in the root (endosphere) and on the root surface (rhizoplane). Thus, the mycobiome in these two compartments was jointly used as the root mycobiome. Overall, the experiment resulted in 45 fungal samples, five time-points × three compartments × three replicates, which were stored at −80 °C until further use. Concomitantly, 10 g of leaves was collected to evaluate the SLA calculated by the ratio of leaf area to leaf dry mass. SLA can indicate the resource utilization strategy of plants and the rate of leaf litter decomposition [23].

### 2.3. Soil Physical and Chemical Property Measurements

Parts of bulk soil were used for measuring the physical and chemical properties. The soil water content (SWC) was measured by soil mass loss before and after drying at 105 °C for 12 h. After the soil was dried, mixed, and sifted using a 2 mm sieve, the soil organic matter (SOM), total N (TN), total P (TP), total potassium (TK), available N (AN), available P (AP), available K (AK), electrical conductivity (EC), and pH were measured according to Zhang et al. [19]. Briefly, SOM was measured by the potassium dichromate oxidation method. The TN content was determined via Kjeldahl digestions. The TP and TK contents were measured after digestion in concentrated HNO_3_ with inductively coupled plasma optical emission spectrometry (iCAP 6300, Thermo Elemental, San Jose, CA, USA). The alkaline hydrolysis method was applicated to measure the AN content. The AP level was measured colorimetrically with the ascorbic acid/molybdate method after extracting using HCl/NH_4_F. The AK was extracted via NH_4_OAc and measured by a flame photometer (FP910, PG Instruments Co., Ltd., Lutterworth, UK). Soil EC was measured in a 1:5 (*w*/*v*) soil: water mixture using an EC meter (DDSJ-319L, INESA Scientific Instrument Co., Ltd., Shanghai, China). Soil pH was measured in a 1:2.5 (*w*/*v*) soil:water mixture using a pH meter (PHSJ-6L, INESA Scientific Instrument Co., Ltd., Shanghai, China).

### 2.4. Molecular Methods

Different pretreatment methods were used for soil and root samples. Total genomic DNA in bulk and rhizosphere soils (~0.5 g) was extracted using the DNeasy PowerSoil DNA Isolation Kit according to the manufacturer’s instructions (Qiagen, Inc., Venlo, The Netherlands). After grinding with liquid nitrogen, DNA in roots (~0.5 g) was extracted using the DNeasy Plant Maxi Kit following the manufacturer’s instructions (Qiagen, Inc., Venlo, The Netherlands). The extracted DNA was quantified and qualified with the NanoDrop ND-1000 spectrophotometer (Thermo Fisher Scientific, Waltham, MA, USA) and agarose gel electrophoresis, respectively. The primers ITS1F/ITS2 were used to amplify the ITS1 region gene. The PCR reactions contained 2.0 μL of DNA extract, 0.25 μL of Q5 High-Fidelity DNA Polymerase, 5.0 μL of High-Fidelity GC Buffer, 1.0 μL of forward/reverse primer, 5.0 μL of Q5 Reaction Buffer, 2.0 μL of dNTPs, and 8.75 μL of ddH_2_O. The PCR conditions consisted of 98 °C for 30 s, then 28 cycles of denaturation at 98 °C for 15 s, 56 °C for 30 s, 72 °C for 30 s, and a final extension at 72 °C for 5 min. PCR products were quantified and mixed with the Quant-iT PicoGreen dsDNA Assay Kit (P7589, Thermo Fisher) on a microplate reader (BioTek, FLx800, Winooski, VT, USA). Subsequently, Illumina^®^ DNA libraries were prepared by purifying and mixing samples in equal proportions according to their DNA concentration and molecular weight, and then were sequenced via the Illumina MiSeq System platform at Personal Biotechnology Co., Ltd. (Shanghai, China).

### 2.5. Bioinformatics Analyses

QIIME 2 2019.4 was used for microbial bioinformatics analyses (https://qiime2.org/ (accessed on: 20 August 2020)). DADA2 plugin was used to control the quality, denoise, splicing, and removal of chimerism to generate amplicon sequence variants (ASVs) [24]. The Naive Bayes classifier was used to assign the taxonomy to ASVs [25]. To ensure sequence uniformity among all samples, the minimum number (3919) of sequences was used as the depth to filter other samples to generate in a filtered ASV table.

### 2.6. Data Analysis

Statistical analysis was implemented in R4.1.0 [26]. A two-way analysis of variance (ANOVA) was used to evaluate the effects of host plant growth time and compartments on the abundance, and α-diversity (estimated by Chao1). Statistical significance was set at the 1% level and *p* values were adjusted by the false discovery rate (FDR). The least significant difference (LSD) was used to identify sources of difference, which was performed by the LSD.test function in the package agricolae [27]. Pearson correlations between soil properties and SLA and fungal abundance or Chao1 were visualized in heatmaps using the package corrplot and the *p* values were FDR-corrected [28]. The Bray–Curtis ASV-level matrix was used to indicate the community dissimilarity between compartments. The Kruskal–Wallis test was performed to identify the difference in the dissimilarity of the mycobiomes. The influences of host growth time and compartment on the mycobiome were calculated via permutational analysis of variance (PERMANOVA) with 999 permutations performed in the package vegan [29] and were visualized by principal coordinate analyses (PCoA). Spearman’s correlations between the mycobiome and soil properties and SLA were evaluated using the Mantel test with 999 permutations. The reduced model implemented in redundancy analysis (RDA), in the vegan package, was performed to identify the soil factors that significantly influenced the mycobiome composition based on the genus-level communities. Venn diagrams were used to demonstrate the influential effects (R^2^) of these driving factors.

Co-occurrence networks were constructed based on Spearman’s correlation (|correlation coefficient| > 0.7 and FDR-adjusted *p* < 0.001) matrix between ASVs in Gephi (https://gephi.org/ (accessed on: 2 September 2020)). The fast greedy algorithms were used to calculate the modularity of networks [30]. The eigengene of each module was annotated using the FUNGuild data [31] and Pearson’s correlations between the abundance of these module eigengenes and environmental factors were analyzed using the cal_eigen function in the microeco package [32]. To evaluate the topological properties of networks, the number of nodes and edges, average path length, network diameter, clustering coefficient, density, heterogeneity, and centralization were selected using the igraph package [33].

## 3. Results

### 3.1. Soil Properties and Plant Traits

The SOM values in perennial communities were significantly higher than in other samples, whereas the SWC was lowest (*p* < 0.05, Table 1). TN, TK, and AN of 30 d samples were significantly higher than those of others (*p* < 0.05), whereas the TP was the lowest. Overall, 90 d seedlings exhibited a higher SLA and then decreased in the 2 a and perennial stages. AK, AP, and EC were comparably different among all samples (*p* > 0.05).

### 3.2. Community Structure of Root-Associated Fungi among Compartments and Host Growth Times

Mycobiome associated with *A. sparsifolia* roots over different growth times was profiled via sequencing of the ITS1 region, resulting in 1532 unique ASVs. Ascomycota was the dominant taxa among all the samples with an average relative abundance of 67.67% (Figure 1). The average abundance of Ascomycota, Basidiomycota, Mucoromycetes, and Tremellomycetes in the rhizosphere and Glomeromycota, Dothideomycetes, and Eurotiomycetes in the root was significantly different among varying host growth times (*p* < 0.05, Figure 1a,b). Ascomycota, Olpidiomycota, Dothideomycetes, and Eurotiomycetes abundance was significantly different among the bulk soil, rhizosphere, and root compartments (*p* < 0.05, Figure 1 and Appendix A). However, the abundance of the dominant phyla and classes in bulk soils was not influenced by host growth times (*p* > 0.05). Except for EC, significant correlations were identified between soil properties and SLA and the abundance of the dominant phyla (Appendix A). Glomeromycota abundance in three compartments was significantly positively correlated with TN and AN content, whereas it exerted a significantly negative correlation with TP and TK (*p* < 0.05). Eighteen ASVs were consistently present in different host growth times and compartments, dominated by the Sordariomycetes and Agaricomycetes classes (Figure 1c).

Chao1 indices of bulk soil and root samples were significantly influenced by host growth times (*p* < 0.01), whereas no difference was observed in the rhizosphere samples (*p* > 0.05, Figure 2a). Significant correlations between environmental factors and Chao1 were highest in the bulk soil samples, followed by root samples (*p* < 0.01), whereas there was no significant correlation between Chao1 of the rhizosphere samples and environmental factors (*p* > 0.05, Figure 2b). Along the bulk soil–plant continuum, the mycobiome displayed a significantly reduced Chao1 and concomitantly an increased community dissimilarity (*p* < 0.001, Figure 2c and Appendix A). PCoA demonstrated that the root communities were separated with bulk soil samples (based on a 95% confidence interval), whereas the rhizosphere communities overlapped with these two compartments (Figure 2d). PERMANOVA results further showed that the mycobiome markedly differed among three compartments (R^2^ = 18.30%, *p* < 0.001) and five host growth times (R^2^ = 20.80, *p* < 0.001), and their interaction explained 31.45% of the community variation (*p* < 0.001).

### 3.3. The Relationships between the Root-Associated Mycobiome and Environmental Factors

Along the soil–plant continuum, an increasing number of factors were found with significant effects on the mycobiome (*p* < 0.05, Figure 3a). SWC, SOM, and SLA were common factors driving variations in the *A. sparsifolia* root-associated mycobiome composition. Furthermore, results of RDA in combination with the reduced model indicated that the soil factors significantly influenced the community composition (*p* < 0.05, Figure 3b and Appendix A). In bulk soil and root samples, SWC significantly distinguished the mycobiome at the perennial stage from that at other stages (i.e., 30 d, 60 d, 90 d, and 2 a) (*p* < 0.05). In the rhizosphere and root compartments, SOM significantly differentiated the mycobiome at the perennial stage from that at the others (*p* < 0.05). Additionally, in the rhizosphere samples, AN was the most influential factor contributing to the changes in the mycobiome at the 30 d stage (*p* < 0.05).

### 3.4. Co-Occurrence Networks

Co-occurrence network analysis was used to explore the influences of compartments and host growth times on interactions of the root-associated mycobiome (Figure 4, Appendix A). Ascomycota and Basidiomycota were dominant in these networks. The root network had a high number of nodes (215) and edges (1875), average degree (17.4), average path length (1.13), density (0.082), heterogeneity (0.92), and centralization (0.13) (Appendix A). The network of perennial samples had a lower number of nodes (181) and edges (632), average degree (6.98), clustering coefficient (0.91), and density (0.04), but a higher average path length (1.32) than the other networks (Appendix A). The network of 90 d seedlings had a lower diameter (2), heterogeneity (0.52), and centralization (0.05) than the other networks. The functional profiles associated with the trophic mode and guild in modules of these networks were identified (Figure 4b). A high percentage of saprotroph genes were found in modules of three compartments, whereas pathotroph and animal pathogen genes were enriched in the root network. There was a low abundance of genes associated with arbuscular mycorrhizal, plant parasite, leaf saprotroph, and bryophyte parasite in the bulk soil and rhizosphere communities. AN, EC, SWC, SOM, and SLA had significant correlations with functional redundancy in communities (Figure 4c).

## 4. Discussion

Significantly compartmental shifts of the mycobiome were observed along the soil–plant continuum, specifically a decreased α-diversity and an increased community dissimilarity (Figure 1, Figure 2 and Appendix A). These findings reinforced the exclusionary role of the root compartment and suggested that host selection plays a dominant role in the formation of *A. sparsifolia* root-associated fungal mycobiomes. Similar observations were observed in the root-associated microbiome of *Arabidopsis thaliana* [34] and other crops including wheat and maize [23,35]. Such findings may be due to the combination of root exudation, host innate immune system, and microbe–microbe interactions, which drives the formation of root fungal community and differentiates it from rhizosphere and bulk soil communities [36]. Moreover, we identified significant differences at the ASV level and some dominant taxa among the five tested times (Figure 1, Figure 2 and Figure 3), suggesting an age-sensitive host selection. Our sampling scheme characterized the developmental patterns of the root-associated mycobiome over various host growth periods. Perennial communities of three compartments were significantly different from other times. The rhizosphere fungus progressed over the first 30 days following germination but then stabilized in composition from 60 to 455 days. Some microorganisms enter plants by expressing (horizontally transferred) genes encoding host cell wall degradation and biosynthesis of secondary metabolites [36]. Additionally, cell wall characteristics change with plant growth, which may be an important aspect of plant shaping its root-associated mycobiome. These findings support our hypotheses about the restructuring of the root-associated mycobiome with the host growth processes.

Despite significant differences in the mycobiome structure between different compartments during various growth times, core taxa were still identified comprising 18 ASVs (Figure 1c). Sordariomycetes, one of the largest classes in the Ascomycota, is involved in the decomposition of hydrocarbon pollutants, nitrous oxide (N_2_O) production, humic acid modifications, and nutrient cycling [37]. The degradation of lignin is primarily accomplished by members of the Agaricomycetes group [38]. Furthermore, our data suggested that three compartments contained a high percentage of ASVs carrying functional genes of saprotrophs (Figure 4b). These results suggest that the core taxa interacted with their hosts in a stable and long-term manner, which may reflect their potential role in facilitating host nutrient acquisition.

Many researchers have pointed out that the availability of soil water and nutrient largely shape the bulk soil microbial communities [9,39]. Our study further reveals that the rhizosphere and root communities were more strongly affected by edaphic factors than the bulk soil community (Figure 3a). A likely explanation for this finding is that the nutrient-poor and stressed bulk soils may induce a community dominated by dormant taxa that were insensitive to environmental changes and the abundance of dominant taxa was slightly affected by the sampling time (Figure 1a,b). In contrast, root deposition regulated by plant physiology creates a nutrient-rich rhizosphere environment [40]. Changes in soil nutrients may indirectly alter the rhizosphere mycobiome by influencing the microbial substrates. The mycobiome inhabiting the root surface and the endosphere are subsets of the rhizosphere community [5] and are regulated by the host immune system [36]. Therefore, they (collectively considered as roots in this study) are influenced by environmental factors to a higher degree. Plant attributes can also regulate soil microbial community structure by altering the quality of resource inputs via litter, and the decomposition of litter can be well indicated by the specific leaf area [16]. Although there was no significant correlation between the SLA and soil nutrients (Figure 3a), this study suggests that plant nutrient acquisition strategies play an important role in shaping the root-associated mycobiome. Plant species with a higher specific leaf area tend to adopt a strategy of rapid growth [37] and may also have an increased turnover rate of fine roots [41], which may potentially alter the root-associated mycobiome during the host growth and development. This may distinguish the mycobiome associated with the *A. sparsifolia* roots in the perennial from other stages. Taken together, these observations support our hypotheses about a joint effect between the edaphic factors and plant attributes on the root-associated mycobiome.

Network analysis not only provides simple information about the richness and composition of microbial communities but also helps us decipher the complex microbial interactions [39]. *Alhagi sparsifolia* root-associated fungal networks had different topological properties across various compartments and host growth times (Figure 4 and Appendix A), supporting our hypothesis. Variations in the topological properties can be explained by niche differences caused by differences in the micro-environment of the compartment among various host growth times. The rhizosphere is one of the most active interfaces on Earth [40]. Root deposits in this habit can attract a variety of microorganisms, leading to complex microbial interactions [39]. However, in our study, the assemblage of the root mycobiome formed a more complex (i.e., increased average degree) and connected (i.e., increased number of nodes and edges) network than the bulk soil and rhizosphere networks, although the root community presented low species richness. For dicotyledons, root tissue is primarily composed of the cortical layer and vascular tissues, and these physical barriers result in a low diversity of the root mycobiome [42]. However, abundant nutrients within roots make the root community active, which may contribute to the complex root network. A complex root network benefits plants by helping their hosts better cope with environmental changes, suppressing soil-borne pathogens, and priming the plant immune system [43,44]. Despite this, a high abundance of genes encoding the pathotroph and animal pathogens still colonized the root (Figure 4b). It was hypothesized that these pathogens may have been inherited from the seed of the parent plants because the tight and complex microbial networks cause difficulty for newcomers (whether beneficial or pathogenic) to invade these niches [45]. This may provide a reference for the prevention and control of animal diseases in local animal husbandry [46]. Additionally, the root fungal network was more heterogeneous (i.e., modularity) than other compartments (Appendix A), which may be related to the heterogeneous and loose substrate architecture of the root tissue [42]. A highly heterogeneous habitat contains high functional redundancy, which can maintain community stability by limiting external disturbances to a certain module, reducing the impact on other parts of the network [47]. Thus, a more heterogeneous root community may contribute to improved root homeostasis in a changing environment.

Plant roots can alter the adjacent soil environment by changing the pH and inputting much C into the soil [48,49]. The specific leaf area reflects the tradeoff between conservative growth and rapid growth strategies for plants [37]. In this study, the specific dynamic leaf area of *A. sparsifolia* was observed among different development times (Table 1), which may exert a dynamic influence on root-associated fungal networks. For example, the conservative growth strategy indicated by a low specific leaf area in the perennial stage may reduce the photosynthetic C allocation to underground or turnover of fine roots, resulting in lower complexity of the root-associated mycobiome in the perennial stage than in other stages.

## 5. Conclusions

Overall, we discovered a significantly varied diversity, composition, and co-occurrence networks of the *A. sparsifolia* root-associated mycobiome between the host developmental stages. Compartment, host developmental stage, and their interaction significantly shaped the root-associated communities. Along the soil–plant continuum, a decrease in α-diversity accompanied by an increase in community dissimilarity indicated a significant host effect. Although the root mycobiome showed low taxonomic diversity, it represented a complex and connected network and was largely influenced by edaphic factors compared to those in the other compartments. The growth strategy of *A. sparsifolia* may have contributed to the difference in community structure and complexity at the perennial stage from other stages. However, future work is needed to uncover the underlying mechanisms of these phenomena. This work may contribute to our understanding of how desert plant–fungi interactions affect the fitness and function of the overall halobiont and increase plant fitness and productivity in the future.

## Figures and Tables

**Figure 1 microorganisms-10-02376-f001:**
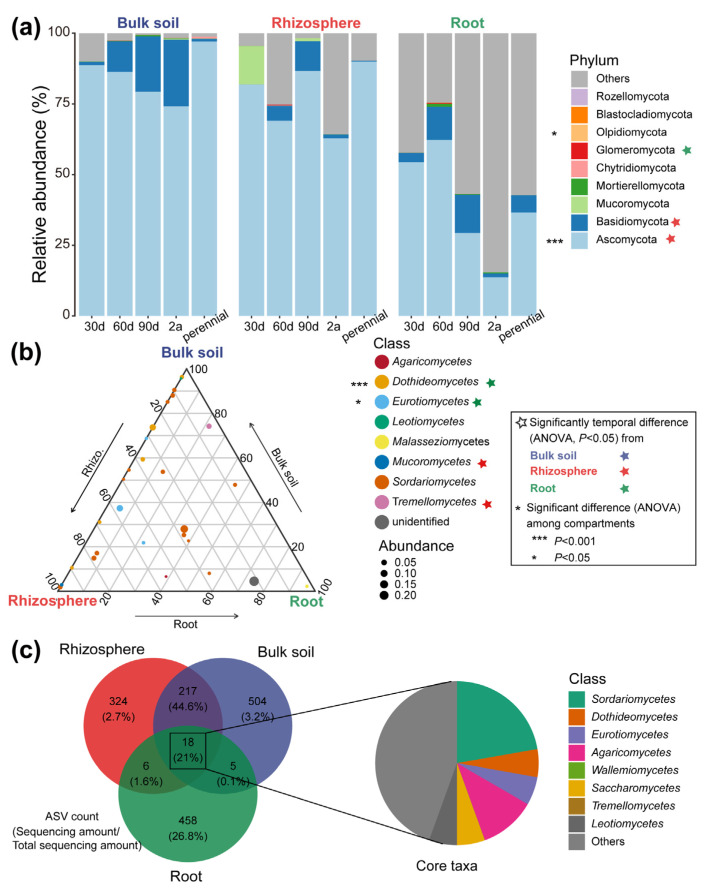
Abundance of *A. sparsifolia* root-associated fungal taxa. (**a**) Average relative abundances at the phylum level. (**b**) Ternary plot indicating the fraction of top 30 genera fractions in each compartment. Circles represent the classes in which these genera belong, and circle size represents the average abundance. (**c**) The Venn diagram shows the number of specific and common ASVs and the proportion of the corresponding sequence to the total sequence. The pie chart shows the composition of these core taxa at the class level. Abbreviations: 30 d, 30 days old; 60 d, 60 days old; 90 d, 90 days old; 2 a, 2 years old; perennial, over 3 years old.

**Figure 2 microorganisms-10-02376-f002:**
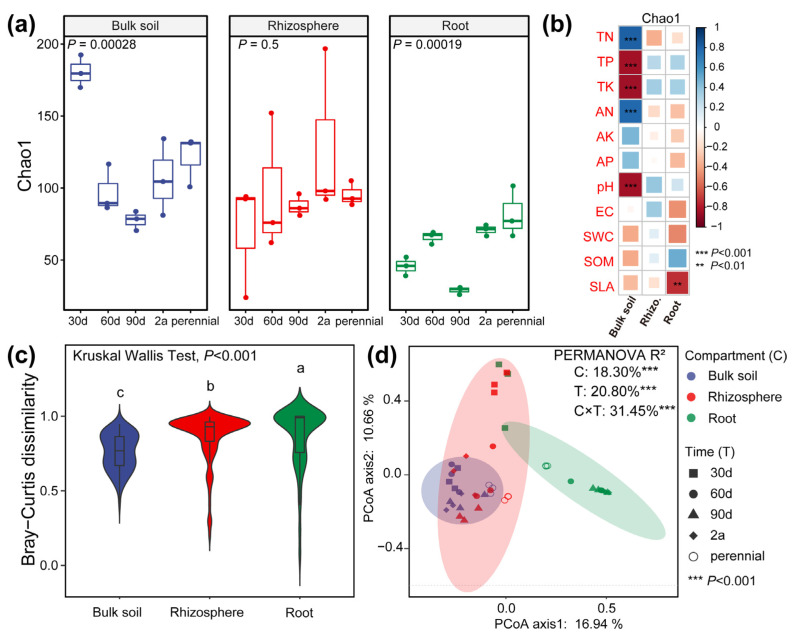
The α- and β-diversity of *A. sparsifolia* root-associated mycobiome. (**a**) Chao 1 of root-associated mycobiome in different host development times. (**b**) Pearson correlations between Chao 1 and soil properties. ***, *p* < 0.001; **, *p* < 0.01. All *p* values were corrected by FDR. (**c**) The Bray-Curtis dissimilarity at the ASV level among three root-associated compartments. Letters indicate significant differences according to the Kruskal Wallis test. (**d**) Principal coordinate analysis (PCoA) in combination with permutational analysis of variance (PERMANOVA). Ellipses indicate a 95% confidence interval. ***, *p* < 0.001. Abbreviations: 30 d, 30 days old; 60 d, 60 days old; 90 d, 90 days old; 2 a, 2 years old; perennial, over 3 years old. TN, total nitrogen (g/kg); TP, total phosphorus (g/kg); TK, total potassium (g/kg); AN, available nitrogen (mg/kg); AP, available P (mg/kg); AK, available potassium (mg/kg); EC, electrical conductivity mS/cm); SWC, soil water content; SOM, soil organic matter (g/kg); SLA, specific leaf area (cm^2^/g). C × T, interaction between compartment and development time.

**Figure 3 microorganisms-10-02376-f003:**
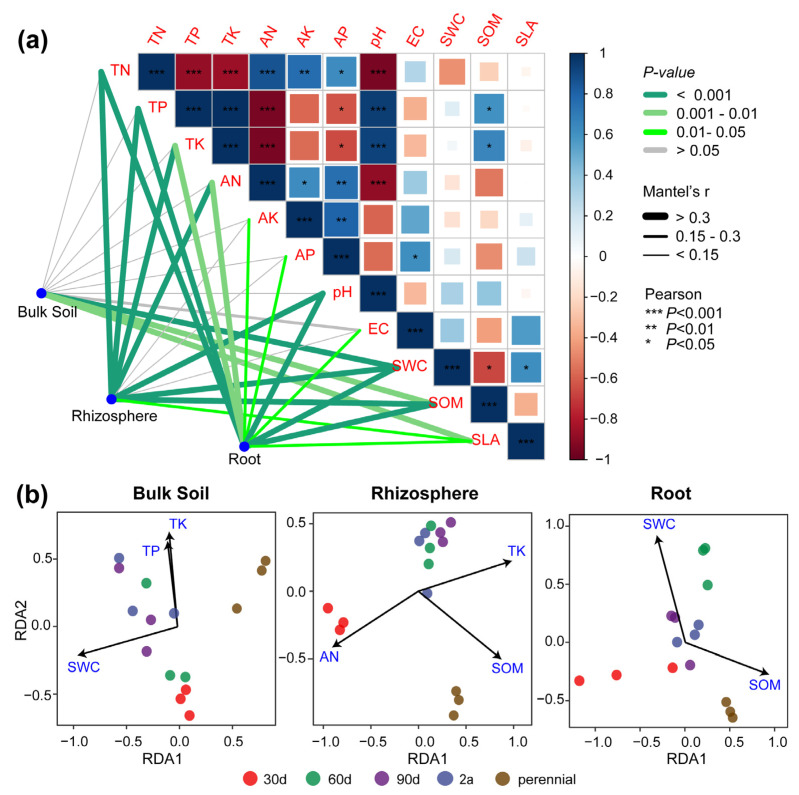
The correlations between the mycobiome and environmental factors. (**a**) Mantel test results show Spearman’s correlations between the mycobiome and environmental factors. The correlation heat map on the upper right shows Pearson’s correlations between environmental factors. (**b**) Redundancy analysis (RDA) results show the relationships between edaphic factors and genus-level community structures tested by 999 permutations. The specific partition of these main edaphic factors is presented in Appendix A. Abbreviations: 30 d, 30 days old; 60 d, 60 days old; 90 d, 90 days old; 2 a, 2 years old; perennial, over 3 years old. TN, total nitrogen (g/kg); TP, total phosphorus (g/kg); TK, total potassium (g/kg); AN, available nitrogen (mg/kg); AP, available P (mg/kg); AK, available potassium (mg/kg); EC, electrical conductivity (mS/cm); SWC, soil water content; SOM, soil organic matter (g/kg); SLA, specific leaf area (cm^2^/g).

**Figure 4 microorganisms-10-02376-f004:**
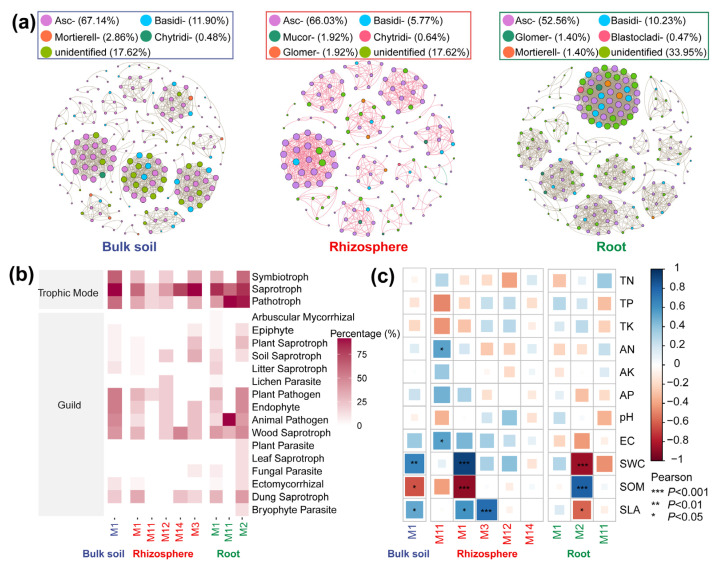
Co-occurrence networks of *A. sparsifolia* root-associated mycobiome. (**a**) Compartment-specific networks. Nodes represent ASVs. Their colors distinguish the phyla they belong to, and their size represents the degree. (**b**) The functional redundancy of fungal co-occurrence network modules among different compartments. (**c**) The Pearson’s correlation between module eigengenes and environmental factors. The M1-M14 represent main modules. Abbreviations: TN, total nitrogen (g/kg); TP, total phosphorus (g/kg); TK, total potassium (g/kg); AN, available nitrogen (mg/kg); AP, available P (mg/kg); AK, available potassium (mg/kg); EC, electrical conductivity (mS/cm); SWC, soil water content; SOM, soil organic matter (g/kg); SLA, specific leaf area (cm^2^/g).

**Table 1 microorganisms-10-02376-t001:** Soil properties and plant traits among different growth times of *A. sparsifolia*
^1^.

Traits	30 d	60 d	90 d	2 a	Perennial
SOM	1.97 ± 0.14 d	3.03 ± 0.04 b	3.14 ± 0.08 b	2.73 ± 0.06 c	4.74 ± 0.03 a
TN	0.23 ± 0.02 a	0.13 ± 0.00 c	0.14 ± 0.00 c	0.13 ± 0.01 c	0.17 ± 0.00 b
TP	0.31 ± 0.01 c	0.60 ± 0.01 a	0.58 ± 0.00 ab	0.58 ± 0.01 b	0.58 ± 0.00 b
TK	21.24 ± 0.55 a	17.82 ± 0.12 c	18.11 ± 0.05 bc	18.11 ± 0.09 bc	18.53 ± 0.19 b
AN	29.87 ± 5.52 a	4.76 ± 0.86 b	4.05 ± 0.24 b	4.76 ± 0.63 b	5.95 ± 0.24 b
AK	143.17 ± 33.77 a	103.00 ± 2.65 a	108.33 ± 7.51 a	90.00 ± 3.61 a	111.00 ± 8.00 a
AP	3.35 ± 0.97 a	2.48 ± 0.14 a	1.74 ± 0.41 a	1.33 ± 0.07 a	1.38 ± 0.08 a
pH	8.09 ± 0.02 b	9.16 ± 0.01 a	9.06 ± 0.04 a	9.23 ± 0.03 a	8.89 ± 0.12 ab
EC	0.43 ± 0.06 a	0.36 ± 0.05 a	0.43 ± 0.06 a	0.33 ± 0.06 a	0.28 ± 0.05 a
SWC	0.06 ± 0.00 c	0.10 ± 0.00 a	0.09 ± 0.00 b	0.08 ± 0.00 b	0.01 ± 0.00 d
SLA	112.16 ± 0.00 b	118.73 ± 6.83 b	144.99 ± 10.03 a	90.01 ± 6.86 c	80.74 ± 5.67 c

^1^ 30 d, 30-day-old; 60 d, 60-day-old; 90 d, 90-day-old; 2 a, 2-year-old; perennial, over 3-year-old. Different lowercase letters in rows indicate significant differences at the *p* < 0.05 level. SOM, soil organic matter (g/kg); TN, total nitrogen (g/kg); TP, total phosphorus (g/kg); TK, total potassium (g/kg); AN, available nitrogen (mg/kg); AP, available P (mg/kg); AK, available potassium (mg/kg); EC, electrical conductivity (mS/cm); SWC, soil water content; SLA, specific leaf area (cm^2^/g).

## Data Availability

The raw sequencing data were deposited at the Sequence Read Archive of the National Center for Biotechnology Information (NCBI), USA (BioProjectID PRJNA797515).

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
