# Peer review of "Alhagi sparsifolia Harbors a Different Root-Associated Mycobiome during Different Development Stages"

_microorganisms, 2022, doi:10.3390/microorganisms10122376_

Round 1
Reviewer 1 Report
Review on manuscript microorganisms-2049335 titled ‘Alhagi sparsifolia harbors a different root-associated mycobiome during different development stages’ by Zhang et al.
I think the subject is interesting for the readers of the MDPI Microorganisms and worth publishing. Nevertheless, a few small changes should be made to the manuscript prior to publication.
Introduction
line 88 typo - Arabidopsis thaliana should be in italics
Results
Table 1. what is 2a? The tables need to be self-explanatory - please add the missing information on abbreviations in the header row.
Figure 1. You plotted only nine phyla. If you follow (as it seems) Tedersoo 2018, there should be 18 fungal phyla. If you divide the grey parts for i.e. 2a in roots among the remaining 9 phyla, those should show abundances higher than that of Basidiomycota. As such I do not think it is appropriate to say that the top nine phyla are plotted. Please add information for grey part as well t.i. plot all fungal phyla.
Figure 2. Description is lacking detailed information. The figures should be self-explanatory. Please provide information what is correlated in b).
Reviewer 2 Report
A well prepared manuscript -- the ease of reading makes such a difference.
raised many questions in my mind and some more discussion on these topics may help other readers too.
Could you spend time adding details to the supplements- these are almost noteform in their brevity
there is much detail I would like to know about
